# FedBPT: Efficient Federated Black-box Prompt Tuning for Large Language Models

## Abstract

Pre-trained language models (PLM) have revolutionized the NLP landscape, achieving stellar performances across diverse tasks. These models, while benefiting from vast training data, often require fine-tuning on specific data to cater to distinct downstream tasks. However, this data adaptation process has inherent security and privacy concerns, primarily when leveraging user-generated, device-residing data. Federated learning (FL) provides a solution, allowing collaborative model fine-tuning without centralized data collection. However, applying FL to finetune PLMs is hampered by challenges, including restricted model parameter access, high computational requirements, and communication overheads. This paper introduces **Fed**erated **B**lack-box **P**rompt **T**uning (FedBPT), a framework designed to address these challenges. FedBPT does not require the clients to access the model parameters. By focusing on training optimal prompts and utilizing gradient-free optimization methods, FedBPT reduces the number of exchanged variables, boosts communication efficiency, and minimizes computational and storage costs. Experiments highlight the framework's ability to drastically cut communication and memory costs while maintaining competitive performance. Ultimately, FedBPT presents a promising solution for efficient, privacy-preserving fine-tuning of PLM in the age of large language models.

## 1 Introduction

Large language models (LLM) have shown increasing power on various NLP tasks (Devlin et al., 2018; Raffel et al., 2020; Brown et al., 2020; Fedus et al., 2022; Zhang et al., 2021; Zeng et al., 2021; Sun et al., 2021; Qiu et al., 2020). Typically, these models are trained on a diverse range of text from books, articles, and websites to gain a broad understanding of human language and are known as the pre-trained language models (PLMs). However, task-specific data is often required to adapt PLMs to perform specific tasks or be more accurate in real-world scenarios. This fine-tuning process relies heavily on user-generated data on devices, providing a wealth of contextual insights and nuanced use cases that reflect actual human interaction and needs. In practice, it is challenging to use these devices and data securely. Data needs to be collected and stored for training, but exchanging and storing sensitive data carries security risks and privacy concerns. To overcome the issue of data isolation, federated learning (FL) can be applied to enable numerous devices to collaboratively finetune PLMs over decentralized data while preserving data privacy (McMahan et al., 2017; Sun et al., 2020).

Although fine-tuning PLMs through FL presents promising opportunities, three challenges constrain their real-world application. Especially for LLMs, these challenges include (1) devices' limited access to the PLM parameters, (2) computational and storage costs for local clients, and (3) communication overhead in the FL system. In the real world, devices utilize LLMs primarily by invoking APIs provided by LLM services (e.g., ChatGPT (OpenAI, 2022; 2023) or NeMo (Kuchaiev et al., 2019)). The clients cannot access the model parameters, thereby being unable to conduct local training. Additionally, even if the clients could access the model parameters, it is impractical for devices with limited resources to conduct local PLM fine-tuning, which is extremely memory-intensive and brings high computational overhead. Moreover, fine-tuning PLMs through FL requires the clients and server to frequently exchange model parameters or gradients, usually in the scale of millions or even billions. Such intensive communication cost is unfeasible for commercial edge devices with limited communication bandwidth. To this end, existing works (Sun et al., 2022a; Chen et al., 2022b; Zhao et al., 2023; Xu et al., 2023) apply parameter-efficient fine-tuning (PEFT) methods of PLMs to FL to

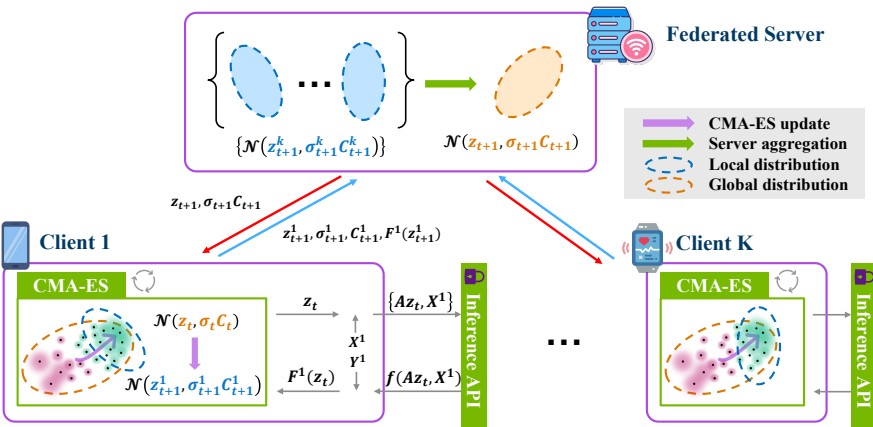

Figure 1: Overview of FedBPT. The clients in FedBPT adopt a gradient-free optimization (CMA-ES) to search for optimal distributions of the prompt based on local data. The clients are not required to access the PLM parameters, and only inference of the PLM is conducted during the search. The server aggregates the uploaded local distributions to derive the globally optimal distribution of the prompt. The global distribution will be sent back to the clients for the next round of search.

reduce resource costs. Effective PEFT methods include adapter tuning (Houlsby et al., 2019), prefix tuning (Li & Liang, 2021), LoRA (Hu et al., 2021) and BitFit (Zaken et al., 2021). These techniques primarily freeze most parameters of PLMs and update only a few additional parameters, which can reduce communication costs significantly. However, these PEFT methods still require the clients to access model parameters and gradients for local training. Even if the computational cost could be reduced, these gradient-based PEFT methods requiring back-propagation are still unfeasible for most edge devices with limited resources, such as mobile phones and AR headsets.

To solve these challenges simultaneously, we propose a new framework called **Fed**erated **B**lack-box **P**rompt **T**uning (**FedBPT**) as shown in Fig. 1. The goal of FedBPT is to train an optimal prompt to improve the performance of the frozen PLMs. The clients and the server exchange prompts rather than model parameters, which reduces the communicated variables from the scale of millions or billions to only hundreds, improving the communication efficiency significantly. The clients in FedBPT adopt a gradient-free optimization method rather than gradient-based methods to conduct local training, which frees the clients from being required to access the model parameters. In addition, only forward-propagation without back-propagation is needed for local training, which can reduce the computational and storage costs for both the devices holding a model and the LLM server that provides inference service APIs.

We conducted experiments on multiple datasets using SOTA PLMs. The results show that FedBPT reduces the communication cost by more than $500k\times$ while achieving comparable results with the baselines that require model parameter access and back-propagation for optimization. FedBPT can also reduce the memory footprint by more than $3\times$ without applying any additional efficient inference technique. By proposing FedBPT, we offer a solution to break down data silos in the era of LLMs without the limiting factors of requiring full model access, large communication bandwidth, and device compute capacity.

We summarize our contributions as follows:

- We present three challenges in applying FL to adapt PLMs in the real world, including the requirement of model access, communication cost, and on-device compute capacity.
- We propose a federated black-box prompt tuning framework (FedBPT) that enables the devices to adapt PLMs in the real world collaboratively by solving the above-mentioned challenges simultaneously.
- We evaluate FedBPT on multiple datasets with SOTA PLMs. FedBPT achieves comparable accuracy with the gradient-based methods that require clients to access model parameters while reducing communication and memory costs significantly.

## 2 RELATED WORKS

### 2.1 FEDERATED LEARNING

Federated learning (FL) (Konečnỳ et al., 2016; McMahan et al., 2017; Sun et al., 2022b) is a prominent distributed learning strategy, particularly beneficial for tasks that prioritize privacy. However, its application faces challenges due to the non-IID nature of distributed datasets. The heterogeneous data distribution across devices compromises accuracy relative to traditional centralized training. Numerous research efforts (Kairouz et al., 2021; Zhao et al., 2018; Chai et al., 2020; Li et al., 2018) have sought to mitigate this performance degradation. Recent works (Chen et al., 2022a; Nguyen et al., 2022) demonstrate that fine-tuning the pre-trained models through FL suffers less from the non-IID issue. Empirical research by Weller et al. (2022) suggests that Pretrained Language Models (PLMs) can diminish the effects of non-IID data and bridge the accuracy discrepancy with centralized training. Their results show that when applying PLMs, even the vanilla FedAvg can achieve comparable model performance with centralized training. These works indicate that FL presents a promising avenue for fine-tuning PLMs by leveraging user data while upholding privacy standards. However, PLMs, especially large-scale ones, introduce considerable communication overheads in FL scenarios, making federated training cumbersome and often unsuitable for practical applications. Additionally, the training of PLMs typically demands ample labeled data to ensure satisfactory accuracy – a condition that may be unattainable for individual device users. It is also noteworthy that many local devices are constrained by limited computational capacity and storage, making the local training of PLMs a challenging endeavor. Diverging from these studies, our work delves into adapting PLMs within FL, especially under tight resource constraints.

### 2.2 PROMPT-BASED LEARNING

Prompt-based learning has gained significant attention in the realm of LLMs. Its essence is rooted in leveraging minimal examples or specific cues to guide a PLM toward the desired output. This contrasts with traditional supervised learning, where a model is trained explicitly using extensive labeled data. OpenAI's GPT-3 (Brown et al., 2020) marked a pivotal turn in the exploration of prompt-based learning. The sheer scale of GPT-3 made it possible to produce relevant outputs with carefully crafted prompts (Brown et al., 2020; Lester et al., 2021) without the need for task-specific model fine-tuning. However, manually designed prompts still suffer a performance gap compared with a fine-tuned model (Brown et al., 2020; Schick & Schütze, 2020; Gao et al., 2020; Sun et al., 2022c). Recent works demonstrate that the prompt does not have to represent natural language. It can also be optimized efficiently in continuous space with gradient descent (Li & Liang, 2021; Hambardzumyan et al., 2021; Qin & Eisner, 2021; Liu et al., 2023; Zhong et al., 2021; Liu et al., 2021). In the case of only tuning the continuous prompt while keeping the parameters of large PLMs untouched, one can retain the efficient training benefits while matching the performance of full model tuning. Prompt tuning (Lester et al., 2021; Li & Liang, 2021) was proposed to fine-tune a continuous vector concatenated to the input embeddings. Unlike manual prompt design conducted at the vocabulary level, prompt tuning optimizes the prompt in the embedding space. Based on this idea, p-tuning (Liu et al., 2021; 2022; 2023) was proposed to improve the performance further. Similar to prompt tuning, p-tuning also learns concrete prompts in the embedding space. However, in p-tuning, an additional LSTM model is required to predict token embeddings.

## 3 PRELIMINARY: BLACK-BOX PROMPT TUNING

Common language understanding tasks can be formulated as a classification task to predict for a batch of input texts $X$ the labels $Y$. Prompt tuning is to train a continuous prompt vector $\boldsymbol{p} \in \mathbb{R}^D$ such that the prediction performance can be improved when the model is fed the optimal prompt vector $\boldsymbol{p}^*$ together with the input $X$. The objective of prompt tuning can be formulated as

$$\boldsymbol{p}^* = \arg\min_{\boldsymbol{p} \in \mathcal{P}} \mathcal{L}\left(f(\boldsymbol{p}; X), Y\right), \tag{1}$$

where $f(\cdot)$ is the PLM inference API, $\mathcal{L}(\cdot)$ is the loss function and $\mathcal{P}$ is some search space of interest. To optimize $\boldsymbol{p}$, gradient-based methods (e.g., SGD) can be applied by conducting back-propagation of the model $f$. Recently, a gradient-free optimization, **B**lack-**B**ox **T**uning (BBT) (Sun et al., 2022d),

was also proposed to optimize the prompt $p$ without back-propagation. Based on the observation that large-scale PLMs have a low intrinsic dimensionality Aghajanyan et al. (2020); Qin et al. (2021), BBT optimizes $z \in \mathbb{R}^d$ in a much smaller subspace ($d \ll D$) and uses a random projection matrix $A \in \mathbb{R}^{D \times d}$ to project $z$ on the original prompt space $\mathcal{P}$. The objective can be formulated as

$$z^* = \arg\min_{z \in \mathcal{Z}} \mathcal{L}\left(f(Az; X), Y\right). \tag{2}$$

To optimize $z$, BBT adopts a gradient-free optimizer CMA-ES (Covariance Matrix Adaptation Evolution Strategy) (Hansen, 2016), a widely used evolutionary algorithm for non-convex black-box optimization in the continuous domain. CMA-ES maintains a parameterized search distribution, i.e., a multivariate normal distribution. In each iteration, CMA-ES samples a population of new query solutions from the multivariate normal distribution as

$$z_{t+1,i} \sim m_t + \sigma_t \mathcal{N}\left(0, C_t\right), \tag{3}$$

where $i = 1, ..., \lambda$ and $\lambda$ is the population size. $m_t \in \mathbb{R}^d$ and $C_t \in \mathbb{R}^{d \times d}$ are the mean vector and covariance matrix of the search distribution at iteration step $t$, respectively. $\sigma_t$ is the standard deviation that controls the step length. $m_t, C_t$ and $\sigma_t$ are updated by maximizing the likelihood of successful steps, which are the steps with lower loss values (cf. Hansen (2016) for more details).

## 4 METHOD

To solve the challenges of model access, communication cost, and computational cost simultaneously, we propose **Fed**erated **B**lack-box **P**rompt **T**uning method (FedBPT) to train an optimal prompt in a federated fashion by adapting BBT to federated learning. Unlike FL methods communicating model parameters, the clients in FedBPT train and communicate with the server prompts rather than the model parameters, which is communication efficient. To optimize prompts, the clients only need to conduct inference rather than back-propagation, significantly reducing the computational cost and memory usage. The FL server aggregates the local prompts uploaded by the client and is completely agnostic to the employed LLM architecture. During training, the clients can treat the model as a black box: neither the clients nor the server requires access to the PLM parameters.

### 4.1 PROBLEM FORMULATION

Suppose there are $K$ clients in FL, and each client hosts a private dataset $D^k = (X^k, Y^k)$ consisting of $n^k$ samples $\{x_i^k, y_i^k\}_{i \in [n^k]}$. Given a global projected matrix $A$ in Eq. (2), the clients collaboratively train an optimal $z$ with the objective to solve:

$$z^* = \arg\min_z \sum_{k \in [K]} \frac{n^k}{\sum_{k \in [K]} n^k} F^k(z), \tag{4}$$

where $F^k(z)$ is the loss of client $k$:

$$F^k(z) = \mathcal{L}\left(f(Az; X^k), Y^k\right) = \sum_{i \in [n^k]} \mathcal{L}\left(f(Az; x_i^k), y_i^k\right). \tag{5}$$

### 4.2 OVERVIEW OF FEDBPT

In FedBPT, the clients optimize local objectives based on BBT. Thus, unlike previous FL works, FedBPT aggregates the CMA-ES parameters applied by the clients to conduct BBT rather than the deep learning models. At the start of the training, the server initializes and distributes the projection matrix $A$ to the clients. Then, the server and clients will freeze and apply $A$ to calculate the prompt with the received $z$. In each communication round (e.g., the $t$-th round), the server first sends the up-to-date global CMA-ES parameters, including the mean vector $z_t$, covariance matrix $C_t$ and the search step $\sigma_t$ to clients. Then, the clients (e.g., the $k$-th client) conduct BBT to optimize the received CMA-ES parameters by minimizing their local loss, i.e. Eq. (5). After local optimization, the clients upload their locally optimal parameters and the local loss value $F^k(z_{t+1}^k)$ to the server. After the

server receives all CMA-ES parameters, it aggregates the local parameters and updates the global CMA-ES parameters for the next communication round. After the training is completed (e.g., $T$ communication rounds), the mean vector of the global CMA-ES $z_T$ will be adopted to compute the optimal prompt $p_T = Az_T$.

The primary distinction between FedBPT and earlier FL algorithms lies in the use of BBT for optimization. Yet, integrating BBT into FL algorithms, such as FedAvg, is not straightforward. Simply combining BBT and FedAvg cannot achieve decent performance. The first challenge is the prompt overfitting problem caused by data distribution shifts across clients, which is common under non-IID settings. The second challenge is how to aggregate CMA-ES parameters on the server effectively. Unlike aggregating deep learning models, directly averaging CMA-ES parameters, mostly consisting of distribution statistics, is not feasible. We will introduce these challenges in detail and our solutions in the following sections.

### 4.3 SERVER-LEVEL CMA-ES ALGORITHM

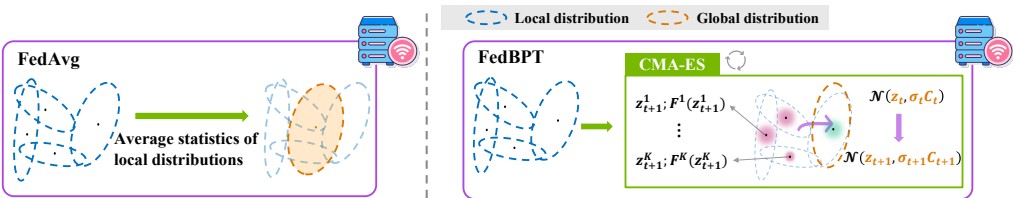

Figure 2: Comparison of aggregation between directly using FedAvg and FedBPT. FedAvg derives the global distribution by directly averaging the local distribution statistics. In FedBPT, the server applies CMA-ES to derive the global prompt distributions with the awareness of the evaluation results of the uploaded local distributions.

After receiving local CMA-ES parameters, the server conducts aggregation on the server to derive a global search distribution that can guide the clients' search in the next communication round. Directly averaging the models uploaded by the clients following FedAvg is not effective for FedBPT. In FedBPT, the clients locally optimize the CMA-ES parameters parameterized by multivariate normal distribution statistics. Directly averaging the standard deviation and covariance matrices via FedAvg cannot derive an optimal global search distribution, as is shown in Sec. 5.2. In addition, CMA-ES is a random search algorithm that cannot guarantee to achieve a local optimum as with gradient-based optimization algorithms. Directly averaging optimal and inferior local search results makes it difficult to achieve a global optimum. To derive an optimal global search distribution on the server, we adopt a server-level CMA-ES algorithm to update the search distribution statistics based on the local search results. The comparison of aggregations by directly applying FedAvg and FedBPT is shown in Fig. 2.

The intuition of the server-level CMA-ES is to consider the local search results as a set of solutions sampled by the server. The server then evaluates these sampled solutions and updates the search distributions for the next communication round. Suppose a set of clients $\mathbb{S}_t$ participate in training in the $t$-th communication round. The server-level CMA-ES takes the received mean vectors $\{z_{t+1}^k\}_{k \in \mathbb{S}_t}$ as the sampled solutions and the local loss values $\{F^k(z_{t+1}^k)\}_{k \in \mathbb{S}_t}$ as the corresponding search step loss. To update the CMA-ES parameters, the search step length is required. However, the server-level "sampling" is conducted by multiple local search steps, and the server-level search step length $\sigma_t$ is intractable. Directly applying a local search step length causes the model to diverge. We provide a theoretical explanation for this divergence in Appendix A. We also theoretically derive a corrected search step length $\sigma'_t$ for the server formulated as

$$\sigma'_t = 2\sqrt{\sum_{k \in \mathbb{S}'_t} \sum_{j=1}^{I} \left(\sigma_{t,j}^k\right)^2 / (|\mathbb{S}_t| \cdot \lambda_k)}, \quad (6)$$

where $\mathbb{S}'_t$ is the set of $\frac{|\mathbb{S}_t|}{2}$ clients that upload $z_{t+1}^k$ with the lowest local loss values $F^k(z_{t+1}^k)$. $\sigma_{t,j}^k$ is the step length of client $k$'s $j$-th local search iteration in communication round $t$. $I$ is the number of local search iterations, and $\lambda_k$ is the local search population of client $k$. The derivation can be found in Appendix A.

## 4.4 LOCAL BLACK-BOX PROMPT TUNING AGAINST OVERFITTING

In real life, client data are non-IID distributed, which causes label-skew across clients (Li et al., 2018). The server-level CMA-ES evaluates the clients' search results based on the uploaded local loss values. Such label-skew makes local searches overfitted to local data distributions by achieving low local loss values and makes it difficult for the server to evaluate their performance on the global data distribution. This overfitting issue is more serious when adopting BBT for local training. Gradient-based optimizations (e.g., SGD) incorporate both data and label information into the gradient for updating. In contrast, when using Eq. (2) as the local training objective, BBT modifies the CMA-ES parameters based primarily on how close predictions are to the labels while using the data only indirectly. It is a practical label-skew case in which most of a client's data is distributed in one class (Tang et al., 2022). In this case, a local CMA-ES might learn a prompt that triggers the frozen PLM to generate predictions corresponding to the dominant class, regardless of the input. To demonstrate this issue, we conduct experiments on AG's NEWS (OpenAI), a topic classification dataset with four data classes. We simulate an FL client to train prompts for a pre-trained RoBERTa (Liu et al., 2019) model using BBT. The simulated client holds data following the Dirichlet distribution, commonly applied in previous FL papers (Hsu et al., 2019; Tang et al., 2022) for non-iid setting, and more than 90% of its data are in class one. The confusion matrix evaluated with the prompt trained by this client is shown in Fig. 3. It is shown that all of the data will be classified as class one after applying the prompt trained by this client, which demonstrates the problem of overfitting caused by local BBT.

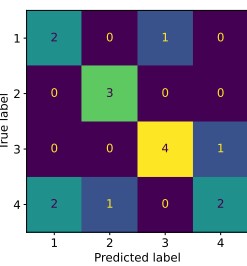

(a) Results without prompt.

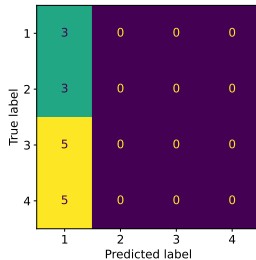

(b) Results of locally trained prompts.

Figure 3: Confusion matrix of a client holding data that more than 90% is in class one.

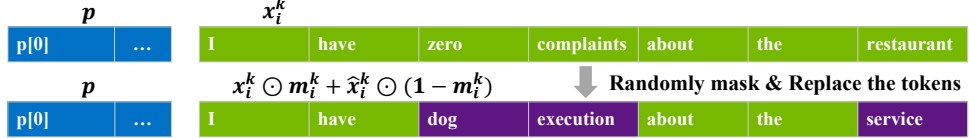

Figure 4: We randomly mask and replace the tokens to perturb a sentence. The PLM should be confused about the perturbed sentence even given an optimal prompt.

To mitigate this overfitting issue, we propose a perturbation method to regularize the local training objective and avoid CMA-ES selecting overfitting prompts. For a sample $\{x_i^k, y_i^k\}$ of client $k$, we randomly generate a binary mask $m_i^k$ with an artificial rate $r_p$ of elements that are zeros. We then randomly sample a sentence $\hat{x}_i^k$ from the vocabulary with the same length of $x_i^k$ as shown in Fig. 4, and the local training objective for the $k$-th client is formulated as

$$z^* = \arg\min_{z \in \mathcal{Z}} \sum_{i \in [n^k]} \frac{\mathcal{L}\left(f(\boldsymbol{A}z; x_i^k), y_i^k\right)}{\mathcal{L}\left(f\left(\boldsymbol{A}z; x_i^k \odot m_i^k + \hat{x}_i^k \odot \left(1 - m_i^k\right)\right), y_i^k\right)}. \quad (7)$$

The intuition is that given a perturbed input, the PLM should not be confident of generating a correct prediction even when fed an optimal prompt.

Applying server-level CMA-ES and local perturbance method, the detailed algorithm of FedBPT can be found in Appendix B.

## 5 EXPERIMENTS

### 5.1 EXPERIMENTAL SETUP

**Datasets and Models** We conduct experiments on three language understanding datasets: (1) The SST-2 (Socher et al., 2013) is a popular sentiment analysis dataset. The SST-2 dataset consists of

sentences taken from movie reviews along with their corresponding sentiment labels. Each sentence is annotated as either "positive" or "negative" based on the sentiment conveyed. (2) The Yelp polarity (yelp) is another sentiment analysis dataset, which consists of reviews on Yelp along with their corresponding sentiment labels of "positive" or "negative". (3) The AG's News dataset (OpenAI) is a large-scale topic classification dataset for the task of categorizing news articles into one of four predefined topic classes. The dataset is based on the AG's corpus, a collection of news articles from various sources. We evaluate our FedBPT on two PLMs: (1) RoBERTa (Liu et al., 2019) is a variation of the BERT model. It is pre-trained using a variant of the masked language modeling (MLM) objective, whose objective is to predict masked tokens in a given text sequence. In this paper, we apply the version of 356 million parameters. (2) Llama 2 (Touvron et al., 2023) is a SOTA PLM released by Meta, which is a collection of foundation language models ranging from 7 billion to 70 billion parameters. Llama 2 models are trained on 2 trillion tokens and have double the context length than Llama 1. In this paper, we evaluate FedBPT on the model with 7 billion parameters.

**Baselines** We compare our black-box tuning FL framework with several gradient-based and gradient-free methods. For gradient-based methods, we compare with three baselines: (1) **FedAvg** (McMahan et al., 2017) is the most widely-used algorithm for FL. In FedAvg, the clients fine-tune the whole model and transmit the updated model parameters. (2) **FedPrompt** (Zhao et al., 2023) is the SOTA work of applying FL to adapt the PLM with high communication efficiency. The clients in FedPrompt learn and transmit prompts, which reduces the communication cost significantly. (3) **FedP-tuning** is built on FedPrompt by replacing the local training from prompt tuning to p-tuning (Liu et al., 2022), which is more advanced and proven to achieve higher performance on downstream tasks. For gradient-free methods, we consider three baselines: (1) **Manual Prompt** is adapted following the templates and label words in Appendix C to conduct zero-shot evaluation. (2) **In-context Learning** Following Brown et al. (2020), we randomly select up to 5 training samples and concatenate them with the input texts. (3) **FedAvg-BBT** is a baseline by simply combining BBT (Sun et al., 2022d) and FedAvg. We build this baseline for comparison as part of an ablation study to show the effectiveness of our designed server-level prompt tuning.

**FL setup & Hyperparameters** We follow FedPrompt (Zhao et al., 2023) to design our FL setup. The system has ten clients, and all of the clients participate in training in each round. Considering the real world, where many users possess only a limited amount of labeled data, we conduct experiments under few-shot settings. We randomly select 40 samples for each class to construct a training set $D_{train}$. We conduct experiments in both IID and non-IID settings. For IID settings, we split the training dataset $D_{train}$ evenly. For non-IID settings, we follow previous works to split the data following the Dirichlet distribution parameterized by $\alpha$. We maintain a default setting of $\alpha = 1.0$ throughout our experiments. The initial search step length $\sigma_1$ is 1. We set local iteration $I$ to 8 and the local population $\lambda_k$ to be 5 for all clients.

## 5.2 EXPERIMENTAL RESULTS

| Method | Trainable Params. | SST-2 | | AG's NEWS | | Yelp | |
|---|---|---|---|---|---|---|---|
| | | Acc.(%) IID | Acc.(%) non-IID | Acc.(%) IID | Acc.(%) non-IID | Acc.(%) IID | Acc.(%) non-IID |
| *Gradient-based methods* | | | | | | | |
| FedPrompt | 51K | 90.25 | 85.55 | 87.72 | 85.62 | 91.44 | 91.47 |
| FedP-tuning | 15M | **90.6** | **87.16** | **88.17** | **86.11** | **93.61** | **91.63** |
| FedAvg | 355M | 84.7 | 82.4 | 77.43 | 76.54 | 88.25 | 88.03 |
| *Gradient-free methods* | | | | | | | |
| Manual prompt | 0 | 83.6 | | 75.75 | | 88.37 | |
| In-Context Learning | 0 | 79.7 | | 76.96 | | 89.65 | |
| FedAvg-BBT | 500 | 84.45 | 84.17 | 76.54 | 76.46 | 89.64 | 89.72 |
| *FedBPT* | *500* | *87.16* | *86.47* | *82.36* | *81.03* | *91.12* | *90.8* |

Table 1: Results under both IID and non-IID settings with RoBERTa as the backbone model.

**Results of RoBERTa.** The results when adopting RoBERTa as the PLM are shown in Tab. 1. Compared with the gradient-based methods, FedBPT achieves comparable or even higher accuracy with drastically reduced trainable parameters. Specifically, FedBPT achieves an accuracy of 0.92%

higher than FedPrompt and only 0.69% lower than the best gradient-based baseline FedP-tuning for SST-2 under the non-IID setting. Meanwhile, FedBPT reduces the trainable parameters by more than $100\times$ and $30,000\times$ compared with FedPrompt and FedP-tuning, respectively. The trainable parameters are required to be transmitted in each communication round, which means that FedBPT reduces the communication cost of one device in one round from 120MB to only 4KB compared with FedP-tuning. For AG's News and Yelp, FedBPT can also achieve comparable accuracy under IID and non-IID settings. Notably, FedAvg cannot improve the accuracy under both IID and non-IID settings. This demonstrates that directly fine-tuning LLMs is not feasible in realistic FL settings when the clients hold limited labeled samples. We document the memory usage by one client of different methods on SST-2 in Tab. 2. It is shown that FedBPT can reduce memory costs by more than $3\times$ compared with gradient-based methods.

Compared with gradient-free baselines, FedBPT achieves higher accuracies under IID and non-IID settings for all the datasets. FedBPT achieves accuracies of 2.3%, 4.57%, and 1.08% higher than FedAvg-BBT under non-IID settings for SST-2, AG's News, and Yelp, respectively. It is shown that FedAvg-BBT achieves limited accuracy improvement compared with manual prompts for all the datasets, which demonstrates that simply combining FedAvg and BBT cannot achieve decent performance. The results show that gradient-based methods outperform gradient-free baselines significantly in accuracy, which is expected. However, we should realize that gradient-based methods require model parameter access and conducting back-propagation, which are not always realistic for most cases of FL, and only the gradient-free methods are feasible in many cases.

| Method | Mem. |
|---|---|
| FedPrompt | 5.8 GB |
| FedP-tuning | 6.1 GB |
| FedAvg | 7.2 GB |
| In-context Learning | 2.1 GB |
| FedBPT | 1.8 GB |

Table 2: Memory footprint on SST-2 by applying RoBERTa.

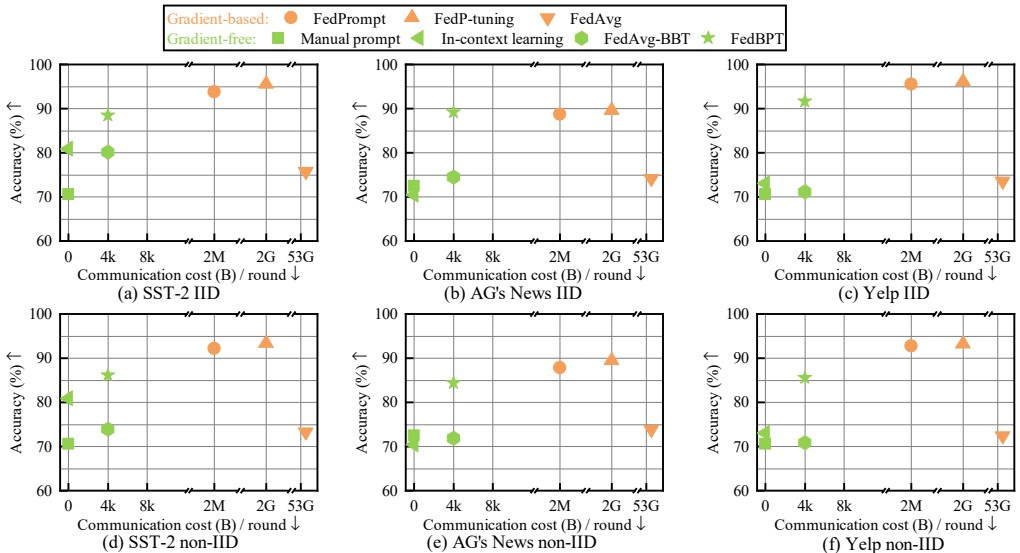

Figure 5: The results under IID and non-IID settings with Llama 2 as the backbone model.

| Method | FedPrompt | FedP-tuning | FedAvg | Manual | FedAvg-BBT | FedBPT |
|---|---|---|---|---|---|---|
| Trainable Params. | 205K | 235M | 7B | 0 | 500 | 500 |

Table 3: Number of trainable parameters when adopting Llama 2 as the backbone model.

**Results of Llama 2.** The number of trainable parameters when applying Llama 2 as the PLM is shown in Tab. 3. The trade-off between the communication cost of one device in one round and model accuracy is shown in Fig. 5. We have three important observations: (1) For Llama 2, FedBPT can improve the accuracy significantly compared with the gradient-free baselines and achieve comparable accuracies with the gradient-based methods in most settings. Specifically, FedBPT improves accuracy

by more than 12%, 11%, and 13% for SST-2, AG's News, and Yelp compared with the manual prompts under non-IID settings, respectively. FedBPT can achieve slightly higher accuracy than FedPrompt under the AG's News IID setting, while the gradient-free baselines experience declines in accuracy of over 15%. (2) FedBPT reduces the number of trainable parameters compared with gradient-based methods even more significantly than adopting RoBERTa. Specifically, compared with FedP-tuning, FedBPT reduces the trainable parameters from 235M to only 500, which means that FedBPT reduces the communication cost of one device in one round from nearly 2GB to 4KB.

In summary, FedBPT can achieve much higher accuracy than gradient-free baselines and comparable accuracy as gradient-based methods for both RoBERTa and Llama 2 models. In addition, the number of trainable parameters does not increase when the model scale is larger. The reason is that FedBPT adopts a projection matrix to project the embedding space to a low-dimension space, which enables the clients to conduct CMA-ES learning to train a low-dimensional vector. This scalability is essential considering the rapid growth of the PLM parameter scale, which allows the clients in FedBPT not to pay more computational or storage costs when the FL system adopts larger PLMs.

### 5.3 ABLATION STUDIES

**Local binary mask rate** ($r_p$). We study the effect of the rate of zeros in the binary masks $m_i^k$ that local devices apply to perturb input and avoid overfitting. We conduct experiments on SST-2 and AG's News under the non-IID setting for RoBERTa. As introduced in Sec. 4.4, a larger $r_p$ means that more tokens in a sentence will be randomly replaced. We set $r_p$ from 0% to 80%, and the results are shown in Tab. 4. It is shown that applying the random placement can improve the global accuracy compared with simply adopting vanilla BBT for local training (i.e., $r_p = 0$). This illustrates the effectiveness of our designed random placement in mitigating the local overfitting challenge.

| Dataset | SST-2 | | | | | AG's News | | | | |
|---|---|---|---|---|---|---|---|---|---|---|
| $r_p$ | 0 | 0.2 | 0.4 | 0.6 | 0.8 | 0 | 0.2 | 0.4 | 0.6 | 0.8 |
| Acc. (%) | 84.86 | 85.21 | 86.03 | **86.47** | 86.12 | 78.28 | 80.92 | **81.03** | 80.75 | 80.83 |

Table 4: Results of FedBPT adopting RoBERTa with different $r_p$ under non-IID settings.

**Local population size** ($\lambda_k$). In each iteration of local search, the clients (e.g., the $k$-th client) sample $\lambda_k$ candidates for evaluation. We study the effect of local population $\lambda_k$ on the model accuracy. We set $\lambda_k$ from 5 to 20, and conduct experiments on SST-2 and AG's News for RoBERTa. The results are shown in Fig. 6. It is shown that the model accuracy of FedBPT is not sensitive to $\lambda_k$. Thus, in real applications, $\lambda_k$ can be set relatively small to reduce computational cost.

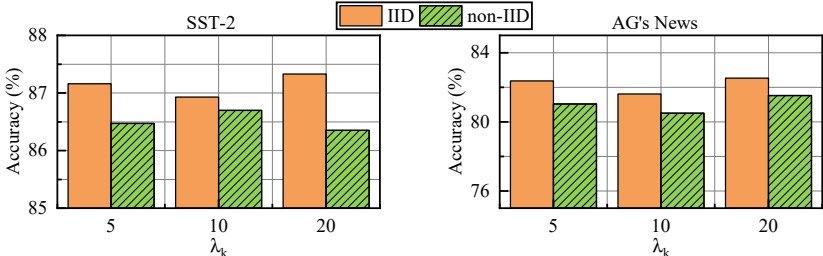

Figure 6: Results of FedBPT adopting RoBERTa with different $\lambda_k$.

## 6 CONCLUSION

We introduced an FL framework, FedBPT, allowing clients to adapt black-box PLMs efficiently using gradient-free optimization. This approach eliminates the need for clients to access model parameters and only requires forward propagation for local training, thus lowering computational and storage demands for devices and LLM service providers. Evaluations of several datasets with SOTA PLMs revealed that FedBPT matches the accuracy of gradient-based methods but with markedly less communication and memory overhead.

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

# A   DERIVATION OF CORRECTED SEARCH STEP ON THE SERVER

The server of FedBPT adopts CMA-ES to conduct a server-level search of the optimal search distribution and search step length. We follow the notation in Hansen (2016) to derive a corrected search step $\sigma'_t$ on the server. After the server receives the locally updated prompt vectors $\{z^k_{t+1}\}_{k \in \mathbb{S}_t}$ and the corresponding loss values $\{F^k(z^k_{t+1})\}_{k \in \mathbb{S}_t}$, the server updates the CMA-ES parameters. Without the loss of generality, we assume that $\left[z^1_{t+1}, ..., z^{|\mathbb{S}_t|}_{t+1}\right]$ is ordered satisfying that $F^1(z^1_{t+1}) < ... < F^{|\mathbb{S}_t|}(z^{|\mathbb{S}_t|}_{t+1})$. The server updates $z_{t+1}$ following

$$
\begin{aligned}
z_{t+1} &= \sum_{k=1}^{\mu} w_k z^k_{t+1} \\
&= z_t + \sum_{k=1}^{\mu} w_k (z^k_{t+1} - z_t),
\end{aligned}
\tag{8}
$$

where we apply *Rank-$\mu$-Update* Hansen (2016) and the $\mu$ best $z^k_{t+1}$ with lowest $z^k_{t+1}$ are summed with the weights $\{w_k\}_{k \in [\mu]}$ to update $z_{t+1}$. There is no issue to update $z_{t+1}$ on the server, but to update $\sigma_{t+1}$ and $C_{t+1}$, an intermediate coefficient *evolution path* value for this round $p_{c,t+1}$ should be derived first. CMA-ES derives the evolution path following

$$
p_{c,t+1} \leftarrow (1 - c_c) p_{c,t} + \sqrt{1 - (1 - c_c)^2} \sqrt{\mu_w} C_t^{-1/2} \frac{z_{t+1} - z_t}{\sigma_t},
\tag{9}
$$

where $c_c$ is an artificial hyper-parameter satisfying $c_c \le 1$, and $\mu_w = \frac{1}{\sum_{k=1}^{\mu} w_k^2}$. As explained in Sec. 4.3, $\sigma_t$ is intractable in FedBPT. Thus, we need to derive an estimated $\sigma'_t$ to conduct CMA-ES on the server correctly. The key to estimating the global search step on the server is to guarantee that the term $\sqrt{\mu_w} C_t^{-1/2} \frac{z_{t+1} - z_t}{\sigma_t}$ follow a standard normal distribution

$$
\sqrt{\mu_w} C_t^{-1/2} \frac{z_{t+1} - z_t}{\sigma_t} \sim \mathcal{N}(0, I)
\tag{10}
$$

under neutral selection, which means that the server randomly selects $z^k_{t+1}$ to update the CMA-ES parameters. Based on this rule of estimation, we first derive the distribution of $z_{t+1} - z_t$. From Eq. (8), we have

$$
z_{t+1} - z_t = \sum_{k=1}^{\mu} w_k (z^k_{t+1} - z_t).
\tag{11}
$$

$z^k_{t+1}$ is formulated as

$$
z^k_t = m_t + \sum_{j=1}^{I} \sigma^k_{t,j} \times \sum_{i=1}^{\mu_l} \alpha^k_{t,j,i} z^k_{t,j,i} \sim \mathcal{N}\left(0, C^k_{t,j}\right),
\tag{12}
$$

where $I$ is the number of local iterations in one round of training, and $\mu_l$ is the rank of local *Rank-$\mu$-Update*. $\sigma^k_{t,j}$ is the search step length of client $k$'s $j$-th iteration in round $t$. $z^k_{t,j,i}$ is the $i$-th sampled point in client $k$'s $j$-th iteration of search in round $t$, and $\alpha^k_{t,j,i}$ is the corresponding weight when conducting weighted sum of $z^k_{t,j,i}{}_{i \in [\mu_l]}$. When the clients conduct limited iterations of local training in one round, we make an assumption that the local covariance matrix $C^k_{t,j}$ in one round will not change significantly, which means that $C^k_{t,j} \approx C^k_{t,1}$, then we have

$$z_{t+1}^k = z_t + \sum_{j=1}^{I} \sigma_{t,j}^k \times \sum_{i=1}^{\mu_l} \alpha_{t,j,i}^k z_{t,j,i}^k \sim \mathcal{N}\left(0, C_{t,j}^k\right)$$

$$\approx z_t + \sum_{j=1}^{I} \sigma_{t,j}^k \times \sum_{i=1}^{\mu_l} \alpha_{t,j,i}^k z_{t,j,i}^k \sim \mathcal{N}\left(0, C_{t,1}^k\right) \tag{13}$$

$$= z_t + \sum_{j=1}^{I} \sigma_{t,j}^k \times \sum_{i=1}^{\mu_l} \alpha_{t,j,i}^k z_{t,j,i}^k \sim \mathcal{N}\left(0, C_t^k\right).$$

Therefore, we have

$$\frac{C_t^{-1/2}\left(z_{t+1} - z_t\right)}{\sqrt{\sum_{k=1}^{\mu} (w_k)^2 \sum_{j=1}^{I} \left(\sigma_{t,j}^k\right)^2 \sum_{i=1}^{\mu_l} \left(\alpha_{t,j,i}^k\right)^2}} \sim \mathcal{N}\left(0, I\right). \tag{14}$$

In this paper, we focus on the CMA-ES algorithm setting that $w_1 = ... = w_\mu = \frac{1}{\mu}$ and $\alpha_{t,j,1}^k = ... = \alpha_{t,j,\mu_l}^k = \frac{1}{\mu_l}$. Hence, we have

$$\frac{C_t^{-1/2}\left(z_{t+1} - z_t\right)}{\sqrt{\sum_{k=1}^{\mu} (w_k)^2 \sum_{j=1}^{I} \left(\sigma_{t,j}^k\right)^2 \sum_{i=1}^{\mu_l} \left(\alpha_{t,j,i}^k\right)^2}}$$

$$= \sqrt{\mu_w} \frac{C_t^{-1/2}\left(z_{t+1} - z_t\right)}{\sqrt{\sum_{k=1}^{\mu} \sum_{j=1}^{I} \left(\sigma_{t,j}^k\right)^2 / (\mu \cdot \mu_l)}} \sim \mathcal{N}\left(0, I\right). \tag{15}$$

Compared with Eq. (10), we derive an estimated global search step length as

$$\sigma_t' = \sqrt{\sum_{k=1}^{\mu} \sum_{j=1}^{I} \left(\sigma_{t,j}^k\right)^2 / (\mu \cdot \mu_l)}. \tag{16}$$

In this paper, we set $\lambda_1 = ... = \lambda_K$ and $\mu_l = \frac{\lambda_k}{2}$, where $\lambda_k$ is the local population size of the $k$-th client, and we set $\mu = \frac{|\mathbb{S}_t|}{2}$. Then without the restriction on the order of $\left[z_{t+1}^1, ..., z_{t+1}^{|\mathbb{S}_t|}\right]$, we have

$$\sigma_t' = 2\sqrt{\sum_{k \in \mathbb{S}_t'} \sum_{j=1}^{I} \left(\sigma_{t,j}^k\right)^2 / (|\mathbb{S}_t| \cdot \lambda_k)}, \tag{17}$$

where $\mathbb{S}_t'$ is the set of $\frac{|\mathbb{S}_t|}{2}$ clients that upload $z_{t+1}^k$ with the lowest local loss values $F^k(z_{t+1}^k)$.

From Eq. (15), we can see that if we directly apply the local search step length $\sigma_{t,j}^k$ as the global search step size, $z_{t+1} - z_t$ will be normalized to follow a normal distribution with a larger variance. In this case, the evolution path cannot be correctly adapted, and the CMA-ES algorithm cannot search an optimal distribution for the global prompt and derive an optimal search step length for the next communication round, which leads to the divergence of the local models.

# B   ALGORITHM

---

**Algorithm 1** Training Algorithm of FedBPT.

---

**Server executes:**
    initialize the projection matrix $\boldsymbol{A}$ and distribute it to the clients
    initialize the global CMA-ES parameters $\{\boldsymbol{z}_0, \sigma_0, \boldsymbol{C}_0\}$
    **for** each round $t = 0, 1, \ldots$ **do**
        **for** each client $k \in S_t$ **in parallel do**
            $\{\boldsymbol{z}_{t+1}^k, \{\sigma_{t,j}^k\}_{j \in [I]}, F^k(\boldsymbol{z}_{t+1}^k)\} \leftarrow \text{ClientUpdate}\,(\boldsymbol{z}_t, \sigma_t, \boldsymbol{C}_t)$
        **end for**
        $\sigma_t' = 2\sqrt{\sum_{k \in \mathbb{S}_t'} \sum_{j=1}^I \left(\sigma_{t,j}^k\right)^2 / \left(|\mathbb{S}_t| \cdot \lambda_k\right)}$
        $\{\boldsymbol{z}_{t+1}, \sigma_{t+1}, \boldsymbol{C}_{t+1}\} \leftarrow \text{CMA-ES}\Big(\big\{\boldsymbol{z}_{t+1}^k, F^k\left(\boldsymbol{z}_{t+1}^k\right)\big\}_{k \in S_t}; \boldsymbol{z}_t, \sigma_t', \boldsymbol{C}_t\Big)$
    **end for**
**ClientUpdate($\boldsymbol{z}_t, \sigma_t, \boldsymbol{C}_t$):**
    $\boldsymbol{z}_{t,1}^k, \sigma_{t,1}^k, \boldsymbol{C}_{t,1}^k \leftarrow \boldsymbol{z}_t, \sigma_t, \boldsymbol{C}_t$
    **for** each local iteration $j$ from 1 to $I - 1$ **do**
        Randomly sample a set of binary masks $M_j^k$ with the same shape of $X^k$ with a rate $r_p$ of elements that are zeros
        Randomly sample a set of tokens $\hat{X}^k$ with the same shape of $X^k$
        **for** $i \in \lambda_k$ **do**
            $\boldsymbol{z}_{t,j,i}^k \sim \mathcal{N}\left(\boldsymbol{z}_{t,j}, \sigma_{t,j} \boldsymbol{C}_{t,j}\right)$
            $\hat{F}^k(\boldsymbol{z}_{t,j,i}^k) = \mathcal{L}\left(f(\boldsymbol{A}\boldsymbol{z}_{t,j,i}^k; X^k), Y^k\right) / \mathcal{L}\left(f(\boldsymbol{A}\boldsymbol{z}_{t,j,i}^k; X^k \odot M_j^k + \hat{X}^k \odot \left(1 - M_j^k\right)), Y^k\right)$
        **end for**
        $\left\{\boldsymbol{z}_{t,j+1}^k, \sigma_{t,j+1}^k, \boldsymbol{C}_{t,j+1}^k\right\} \leftarrow \text{CMA-ES}\Big(\big\{\boldsymbol{z}_{t,j,i}^k, \hat{F}^k\left(\boldsymbol{z}_{t,j,i}^k\right)\big\}_{i \in [\lambda_k]}; \boldsymbol{z}_{t,j}^k, \sigma_{t,j}^k, \boldsymbol{C}_{t,j}^k\Big)$
    **end for**
    $\boldsymbol{z}_{t+1}^k \leftarrow \boldsymbol{z}_{t,I}^k$
    $F^k(\boldsymbol{z}_{t+1}^k) = \mathcal{L}\left(f(\boldsymbol{A}\boldsymbol{z}_{t+1}^k; X^k), Y^k\right)$
    return $\{\boldsymbol{z}_{t+1}^k, \{\sigma_{t,j}^k\}_{j \in [I]}, F^k(\boldsymbol{z}_{t+1}^k)\}$ to server

---

## C  MANUAL PROMPT TEMPLATES

| Model | Dataset | Template |
|-------|---------|----------|
| RoBERTa | SST-2 | $\langle S \rangle$. It was $[MASK]$ . |
|  | AG's News | $[MASK]$ News: $\langle S \rangle$ |
|  | Yelp | $\langle S \rangle$. It was $[MASK]$ . |
| Llama 2 | SST-2 | What is the sentiment of this sentence: $\langle S \rangle$? $[OUTPUT]$ |
|  | AG's News | $\langle S \rangle$. The news is about $[OUTPUT]$ |
|  | Yelp | $\langle S \rangle$. It is $[OUTPUT]$ |

Table 5: Manual prompt templates. $\langle S \rangle$: sentence in the dataset. $\langle MASK \rangle$: mask token adopted by RoBERTa. $[OUTPUT]$: output placeholder of Llama 2.

We set manual prompts for different datasets as shown in Tab. 5. We set different prompts for RoBERTa and Llama 2 according to their pre-training strategy. For RoBERTa pre-trained on masked language modeling (MLM) tasks, we apply manual prompts to organize the datasets as MLM datasets. For Llama 2, which is pre-trained on causal language modeling (CLM) tasks, we organize the samples to prompt the model to generate labels.

