# OpenReview forum: "FedBPT: Efficient Federated Black-box Prompt Tuning for Large Language Models"
_ICLR.cc/2024/Conference — Submitted to ICLR 2024_

### Official Review · Reviewer_2J6K · 2023-10-31

**Soundness:** 2 fair
**Presentation:** 3 good
**Contribution:** 2 fair
**Rating:** 5
**Confidence:** 3

**Summary:**

This paper discusses three challenges while fine-tuning LLM in federated learning (FL): (i) limited access to the LLM parameters, (ii) computational and storage costs for local clients, and (iii) communication costs between the server and the clients. Upon these challenges, the authors introduce Black-Box Tuning (BBT), a gradient-free approach. It optimizes the prompts via the CMA-ES optimizer for the local update and the global aggregation. To avoid overfitting, the authors propose a perturbation method that mixes up two sentences. The experiments cover three classification tasks and two state-of-the-art models (LLaMA and RoBERTa), and the results indicate the proposed method reduces the trainable parameters, computation overhead, and communication costs.

**Strengths:**

1. It is a promising topic to discuss LLM in FL. This paper is well-written and points out the potential challenges when we fine-tune LLM in FL.
2. The figures of the paper help the reader understand the work clearly.
3. I appreciate the experiments adopting the state-of-the-art pretrained model LLaMA, which shows the method's broad coverage in terms of language models.

**Weaknesses:**

1. As mentioned by the authors, the clients have limited access to the entire model. I agree with this point, but the paper does not work it out. According to Eq. (1), the proposed method should attain the value of $f(\cdot, \cdot)$ to compute the loss function. I think there are only two approaches to the goal: (i) the clients upload the input and the output to the server, and the server computes and returns the loss to the clients; (ii) clients are allowed to use the model to get the loss. The first one breaches the principle requirements of FL, where the clients' data cannot be disclosed to any other parties, while the second one does not match the initial aims because the clients can load the entire model locally.
2. In the experiments, I see the performance degradation between the gradient-based methods and FedBPT. The proposed FedBPT takes advantage of the computation overhead each round because it requires forward propagation only. However, I think the gradient-based methods will take fewer rounds to converge or perform better than FedBPT. As for their high communication overhead, I believe LoRA can solve the issue.
3. Section 3 mentions two dimensions' values (i.e., $d$ and $D$) for the prompt. Intuitively, a longer prompt will perform better. The authors should discuss how these two values affect the model performance.
4. Although Section 3 mentions this paper focuses on the classification task, I think it is too simple because we expect LLM to work on more complicated tasks such as question answering. Also, the datasets used in the experiment are not very challenging because they are binary classification or four-category tasks.

**Questions:**

Please address my concerns in the weaknesses part.

---

> ### Author Response · Authors · 2023-11-16
> **Authors' response**
>
> **W1**. Thanks for the great comment, and we will describe our setting more clearly in the revision. The LLM service providers are not the FL server. The LLM service providers are the APIs (e.g., ChatGPT) that the clients will continue utilizing for inference in the deployment phase. Thus, we have an assumption that if the clients do not have model access, it is acceptable to send only data with a prompt to the LLM service providers for inference, which is realistic since the clients do not have more risk of privacy leakage compared with the deployment phase. In our framework, the clients do not need to send labels to the LLM server, which may contain more sensitive information (e.g., the diagnosis of a patient), and preserve the privacy of the clients’ labels which constitutes an important asset for the clients. The clients use the inference results received from the server and the local labels to compute the loss. Furthermore, our work is not limited to the setting in which the devices have no model access. The LLM APIs are not necessarily on the cloud, our work is also motivated by the setting that the clients are the devices with model access but without enough memory and computational power to compute the back-propagation of LLMs. It is also notable that training DNNs on edge devices is always a problem in real life (e.g., pytorch-mobile and TensorFlowLite still do not support on-device training GPU acceleration). In these settings, the devices can only conduct inference rather than back-propagation, and our FedBPT allows them to still participate in federated finetuning.
>
> **W2**. Thanks for this constructive comment. Evolutionary algorithms are typically slower than SGD under the central training setting. However, in our design, each client plays the role of a “search node”, which improves the parallelism and accelerates the search process. To achieve the accuracies under non-IID settings of SST-2 in Table 1, FedPrompt requires more than 300 communication rounds, while our FedBPT only needs less than 100 communication rounds, which converges in fewer rounds. We will include our clarification in our revision with the assessment of convergence time for all the settings and datasets. There are many techniques to reduce the communication cost of gradient-based methods, but in many settings, gradient-based methods are not feasible, and we can only apply gradient-free methods, which are the focus of our paper. For example, when you are using ChatGPT, you do not have access to the model parameters and can only get the inference results by sending the request to the model through the API provided by OpenAI. Gradient-based methods can achieve relatively higher accuracy, but they are unfeasible in such realistic settings where the devices have no access to model parameters or have no power to conduct back-propagation.
>
> **W3**. The length of the prompt is related to the trade-off between the memory cost and accuracy. It is true that a longer prompt can perform better. However, a longer prompt also causes higher memory costs during inference. We follow previous works [1,2] to set the length of the prompt in our paper. Based on previous works [3], LLMs always have a low intrinsic dimensionality. Thus, we reduce the dimension of optimized parameters to $d$ through a fixed projector to reduce the communication cost of FL and the computational cost of CMA-ES. We will include this discussion in our revision.
>
> **W4**. Even though the datasets are simple, we still outperform other gradient-free methods in terms of accuracy. We agree that the classification task is not too hard in real life, but it is notable that this is the first paper to finetune LLMs in FL using a gradient-free optimization method. We follow existing works [1] on gradient-free optimization of LLMs to set our experiment dataset. It is always challenging to finetune DNNs using gradient-free methods, let alone LLMs. We need to work hard with the whole community to improve the performance and scalability of our proposed framework. However, we believe that we show the community it is a promising direction to apply gradient-free methods in fine-tuning LLMs through FL, which pushes the forefront of deploying FL in real life significantly. Notably, LLMs are becoming larger, and the clients are less and less able to perform gradient-based optimization on devices (even larger ones). The newer and better pre-trained LLMs would be more generalizable to prompt learning, which makes our framework more promising. This concern may come from the lack of limitation discussion in the paper. We will include this limitation and future works in the revised version.
>
>
> [1] Sun, Tianxiang, et al. "Black-box tuning for language-model-as-a-service." ICML2022.
>
> [2] Li, Xiang Lisa, et al. "Prefix-tuning: Optimizing continuous prompts for generation."
>
> [3] Armen Aghajanyan, et al. "Intrinsic dimensionality explains the effectiveness of language model fine-tuning."

---

> > ### Comment · Reviewer_2J6K · 2023-11-21
> >
> > Thanks for your response. Regarding your response, I have some follow-up comments:
> >
> > **C1.** In my opinion, a client's local data is solely accessed on its own. Therefore, even if the LLM holder does not take the role of FL server, the user's data or query cannot be accessed by others. Follow your example of a hospital. A patient's details and symptoms are more sensitive than the diagnostic results.
> >
> > Assuming the input data $X_k$ are not very sensitive, there is another challenge. As we know, the output of LLM $f(Az, X_k)$ is a matrix, meaning the communication cost is critical. Besides, most LLM APIs simply return the generated results. I wonder how the loss is calculated.
> >
> > **C2.** I agree that the proposed work has better performance in terms of gradient-free methods in those simple classification tasks. However, users leverage LLM to solve more complex problems, such as math problem-solving (e.g., GSM-8K). It will be a great contribution if the proposed work can perform well in such a complex task.
> >
> > As I don't see any new results in your latest revision, I cannot justify the arguments in **W3**. To sum up, I will keep my current rating.

---

> > > ### Author Response · Authors · 2023-11-22
> > > **Thanks for the reviewer's reply**
> > >
> > > Thanks for your reply.
> > >
> > > C1. We do not cause additional privacy leakage compared with the deployment phase when the clients send their data to the LLM API for inference. The only extra thing the clients need to share to participate in training is the learned prompt, just like the updated parameters in FedAvg. We think it is a common case nowadays and in the future that the clients conduct inference through the LLM APIs. Notably, FL is a rapidly evolving field. Nowadays, the concept of FL has become more and more general and includes different paradigms. For example, vertical federated learning (VFL) allows all the clients to update the embedding of the data to an active client or a server, which is acceptable to the FL community. In addition, if the reviewer still thinks it is impractical to use LLM APIs in real life, our work can also be applied to the scenario where the devices only have computational power for inference rather than back-propagation of the LLMs, which would allow them to keep the input data on the device.
> > >
> > > In our work, the output of LLM $f(Az, X_k)$ is the logits of prediction. The LLM APIs are not necessarily the chatbot. For example, ChatGPT allows the users to access the embeddings of the output through an API(https://platform.openai.com/docs/guides/embeddings). It is not hard to calculate the logits with the embeddings. We believe that more types of LLM APIs will be released in the future.
> > >
> > > C2. We follow existing works [1] on gradient-free optimization of LLMs to set up our experiments. We will leave the more complex tasks for future work. We still want to emphasize that no existing works can work in the practical setting of our paper, and FedBPT is the first successful attempt in this field. We believe that we show the community it is a promising direction to apply gradient-free methods in fine-tuning LLMs through FL, which significantly pushes the forefront of deploying FL in real life.
> > >
> > > We did not upload the revised paper because we did not want to treat the discussion period as a shepherding process for revising the paper. We focus on addressing all the reviewers’ concerns through discussion and prefer revising the paper afterwards by gathering all the comments in the reviewing process. Otherwise, we cannot guarantee the paper is well organized by making piecemeal revisions. For W3, we did not realize that you were expecting to see more empirical results. We think the analysis and the empirical results in the reference papers we provided in the rebuttal can solve your concern to some extent since the length of the prompt is not a focus of our design and we just follow the convention. We are also glad to provide the results on SST-2_noniid_RoBERTa on shorter prompts with 20 tokens. The results of accuracy are shown below:
> > >
> > > | 20 tokens | 50 tokens|
> > > |---------------|-------------|
> > > |81.31|86.47|
> > >
> > > It is shown that shorter prompts will cause performance degradation, which is consistent with our rebuttal. We will include more results in the revision. However, due to the limited time, we can only provide this compared result.
> > >
> > > Feel free to reply if you have additional concerns.
> > >
> > > Best,
> > > Authors

---

### Official Review · Reviewer_uWcP · 2023-11-01

**Soundness:** 2 fair
**Presentation:** 3 good
**Contribution:** 3 good
**Rating:** 5
**Confidence:** 4

**Summary:**

The paper studies black-box prompt tuning in the federated setting and proposes FedBPT. Specifically, FedBPT extends an existing approach BBT to the federated setting, which enables black-box prompt tuning in the centralized setting. Each client locally updates the prompt mapping vector and sends it to the server. The server averages the vectors and sends the global vector back to the clients for the training of the next round. FedBPT adopts a modified search step length and regularization by perturbing the input prompt.

**Strengths:**

1. The studied problem is important. With the fast development of LLMs, prompt tuning has been a popular research direction and how to enable it in the federated setting is emerging.

2. The paper is well-written and easy to understand.

**Weaknesses:**

1. One concern is about the motivation of the paper. The paper assumes that LLMs can only provide API inference such as GPT-4. In such a case, the clients need to send the prompts to the service provider for inference. However, one important aspect of federated learning is privacy, and the data should not be transferred. Thus, the proposed approach is not feasible in the real world.

2. Experiments miss some important baselines. Considering the vector z as a model, existing FL approaches such as FedProx [1] to solve non-IID data may be also applicable. Comparing these approaches should be added.
[1] Federated optimization in heterogeneous networks

3. Another concern is about the privacy risks of transferring vector z. I think the authors can present the learned prompts in the experiments and see the relationship between them and the training data. If the prompts contain information about the training data, transferring vector z may not be a feasible solution as every client can compute the prompt by $Az$.

**Questions:**

1. How to enable inference without transferring the data in the black-box setting?

2. Can you demonstrate the learned prompts in the experiments?

3. Can you add FL baselines on non-IID data by treating z as a model?

---

> ### Author Response · Authors · 2023-11-16
> **Authors' response**
>
> **W1**. Thanks for the great comment. The LLM service providers are not the FL server, which is the APIs (e.g., ChatGPT) that the clients will continue utilizing for inference in the deploying phase. Thus, we have an assumption that if the clients do not have model access, it is acceptable to send only data with a prompt to the LLM service providers for inference, which is realistic since the clients do not have more risk of privacy leakage compared with the deployment phase. We will claim this point more explicitly in the revised version. In our framework, the clients do not need to send labels to the LLM server, which may contain more sensitive information (e.g., the diagnosis of a patient), and therefore preserve the clients’ label privacy which constitutes an important asset for the clients. In addition, the LLM APIs are not necessarily on the cloud, our work is also motivated by the setting that the clients are the devices without enough memory and computational power to compute the back-propagation of LLMs. It is also notable that efficiently training DNNs on edge devices is always a problem in real life (e.g., pytorch-mobile and TensorFlowLite still do not support on-device training GPU acceleration). In these settings, the devices can only conduct inference rather than back-propagation, and our FedBPT allows them to participate in federated finetuning.
>
> **W2**. Thanks for the suggestion to include more baselines. Our goal is to design a framework that enables resource-limited devices without model access to finetune LLM in a federated fashion. For example, when you are using ChatGPT, you do not have access to the model parameters and can only get the inference results by sending the request to the model through the API provided by OpenAI. Gradient-based methods can achieve relatively higher accuracy, but they are unfeasible in such realistic settings where the devices have no access to model parameters or have no power to conduct back-propagation. We conduct experiments primarily to show that we can improve accuracy compared with the other gradient-free methods with extremely efficient communication. Even though it is not our focus, we will include more discussion around these related techniques in our revision.
>
> **W3**. This is a great comment. Actually, there have been many papers [1,2] discussing data privacy leakage through the LLM gradients in FL, but there is no paper claiming that prompt updating can cause data leakage. There is no existing way to evaluate the relationship between the prompts and the training data. We conduct experiments to explore the similarity of the learned prompt embedding $Az$ to the word embeddings in the training data vocabulary. For SST-2_RoBERTa, the maximal cosine similarity between the prompt embedding and the training word embeddings is only 0.22, and the mean cosine similarity is 0.08, which is relatively low. We “translate” the learned prompt embedding on SST-2_RoBERTa to the natural language by replacing the prompt token with the word with the highest similarity in the vocabulary. The translated prompt is **['Ġcould', 'Ġ13', 'Ġpost', 'Ġmillion', 'Ġreal', 'Ġmen', 'Ġsuch', 'h', 'Ġwho', 'ie', 'ĠAmerica', 'Ġtotal', 'Ġ2015', 'z', 'Ġjust', 'ĠBut', 'Ġweeks', 'Ġteam', 'Ġformer', 'Ġopen', 'Ġmorning', 'ĠHouse', 'Ġcomes', 'Ġwomen', 'Ġformer', 'Ġ2015', 'Ġfinal', 'e', '2', '8', 'Ġof', 'Ġoil', 'Ġ9', 'ĠMr', "'m", 'Ġsays', 'Ġbut', ']', 'Ġfuture', 'ak', 'com', 'b', 'Ġsales', 'Ġc', 'ist', 'Ġplay', 'ĠNews', 'b', 'Ġnext', 'ĠS']**. We compared it with the training samples and did not find a clear relation with the input. Thus, we believe that it is safe to transmit the prompt embedding to the FL server and update the global prompt.
>
> **Q1**. A practical example is that the CMA-ES is deployed on the client’s device, and the LLM is deployed on an LLM service provider’s cloud (e.g., OpenAI). The clients conduct inference through the inference API (e.g., ChatGPT) and compute the evaluation results locally based on the inference results received from the API.
>
> **Q2**. Please refer to our response to W3.
>
> **Q3**. FedPrompt is actually a baseline that treats z as a model and applies FedAvg. We will add discussion around this in the revised paper.
>
> [1] Melis, L., Song, C., De Cristofaro, E., and Shmatikov, V. Exploiting unintended feature leakage in collaborative learning. In 2019 IEEE Symposium on Security and Privacy (SP), pp. 691–706. IEEE, 2019.
>
> [2] Fowl, Liam, et al. "Decepticons: Corrupted transformers breach privacy in federated learning for language models." arXiv preprint arXiv:2201.12675 (2022).

---

> > ### Comment · Reviewer_uWcP · 2023-11-22
> >
> > Thanks for your response. Since there is no revised paper provided, my concerns remain. For W1, I suggest you add more clarification about the settings and add more practical applications. I'll keep my score.

---

> > > ### Author Response · Authors · 2023-11-22
> > > **Thanks for the reviewer's reply**
> > >
> > > Thanks for your reply. We did not upload the revised paper because we did not want to treat the discussion period as a shepherding process for revising the paper. We focus on addressing all the reviewers’ concerns through discussion and prefer revising the paper afterwards by gathering all the comments in the reviewing process. Otherwise, we cannot guarantee the paper is well organized by making piecemeal revisions. Feel free to remind us if there is any concern that we have not solved during the rebuttal.
> > >
> > > For W1, a realistic setting is that the clients utilize ChatGPT for their tasks and want to derive an optimal prompt collaboratively to improve the performance of their tasks. However, they do not want to send their labels to OpenAI or share data with the other clients. In this setting, our framework is the only applicable solution. Another realistic setting is that the clients have the LLMs deployed on their devices, but their devices do not have enough computational power and memory to conduct back-propagation. Then our work enables them to collaboratively train an optimal prompt by only conducting inference.

---

### Official Review · Reviewer_juc2 · 2023-11-03

**Soundness:** 2 fair
**Presentation:** 3 good
**Contribution:** 2 fair
**Rating:** 3
**Confidence:** 5

**Summary:**

This paper introduces a federated prompt-tuning framework for large language models. The key difference in this framework is that it does not use gradient-based optimization methods such as SGD to optimize the trainable prompt parameters. Instead, it adopts a gradient-free optimization method, CMA-ES. In each training iteration, clients optimize the trainable parameters using the CMA-ES method and then upload these parameters to the server. The server does not aggregate these parameters directly, as in FedAvg. Instead, it uses these uploaded parameters to perform another round of CMA-ES updates. The updated parameters are treated as aggregated ones and are distributed among clients for the next iteration. Additionally, the paper proposes randomly masking a portion of the input text sequences to prevent prompts from overfitting.

**Strengths:**

1.	The paper provides a derivation of the step size related to the CMA-ES aggregation on the server side, making the method theoretically more reliable.

2.	This method has fewer trainable parameters and lower communication costs than other prompt tuning methods.

3.	The paper has a detailed introduction to gradient-free optimization, which makes the paper easy to follow.

**Weaknesses:**

1.	A main concern is the time efficiency of this method since evolutionary algorithms are typically slower than SGD. If the paper uses the gradient-free optimization CMA-ES instead of SGD, that might result in not only increased time consumption but also heightened computational resource usage, subsequently slowing down the convergence rate. This paper should assess time efficiency and discuss the additional time overhead introduced by CMA-ES.

2.	Aside from incorporating some existing methods, the primary innovation in this paper revolves around the aggregation of CMA-ES parameters within the framework of federated learning. Other components, such as gradient-free optimization or the utilization of sentence perturbation for preventing overfitting, are hard to label as innovative. Hence, the overall technical novelty of the article is limited.

3.	Baseline methods for prompt tuning in the experiments are overly basic and outdated. Comparisons should be made with more recent state-of-the-art approaches, such as LPT [1] and IDPG [2], which all claimed to perform better than vanilla prompt tuning.

[1] Xiangyang Liu, Tianxiang Sun, Xuanjing Huang, and Xipeng Qiu. 2022c. Late prompt tuning: A late prompt could be better than many prompts. In Findings of the Association for Computational Linguistics: (EMNLP), pages 1325–1338. Association for Computational Linguistics.

[2] Zhuofeng Wu, Sinong Wang, Jiatao Gu, Rui Hou, Yuxiao Dong, V.G.Vinod Vydiswaran, and Hao Ma. 2022. IDPG: An instance-dependent prompt generation method. In Proceedings of the 2022 Conference of the North American Chapter of the Association for Computational Linguistics: Human Language Technologies, pages 5507–5521.

**Questions:**

1.	I noticed the authors emphasize that 'this approach eliminates the need for clients to access model parameters.' However, it seems that clients must have the capability to perform inference on their local devices to optimize CMA-ES parameters, which implies they do need to access model parameters.

2.	From Table 1, I noticed that FedAvg (fine-tuning all parameters) performs noticeably worse than the other two prompt tuning methods. Is there a specific reason for this? In most prompt-tuning literature, fine-tuning's performance is either close to or better than prompt tuning.

---

> ### Author Response · Authors · 2023-11-16
> **Authors' response**
>
> **W1**. Evolutionary algorithms are typically slower than SGD under the central training setting. However, in our design, each client plays the role of a “search node”, which improves the parallelism and accelerates the search process. To achieve the accuracies under non-IID settings of SST-2 in Table 1, FedPrompt requires more than 300 communication rounds, while our FedBPT only needs less than 100 communication rounds. CMA-ES algorithm only updates the multivariate Gaussian distribution of the prompt with 500 variables, which is super efficient and can be neglectable compared to model training with SGD. Thanks for this constructive comment. We will include our clarification in our revision with an assessment of convergence time in different settings.
>
> **W2**. Thanks for the comment, and we are glad to introduce the contribution and novelty of our paper. First, we want to claim that we are motivated by a very realistic setting in which the clients use LLM APIs and have no access to the model parameters. For example, when you are using ChatGPT, you do not have access to the model parameters and can only get the inference results by sending the request to the model through the API provided by OpenAI. In this practical setting, no existing FL solution works, and our work is the first successful attempt. Technically, our paper is not simply combining BPT and FedAvg. We show that simply combining BPT and FedAvg cannot achieve good performance, as shown in our results (the “FedAvg-BBT” baselines). We explore in our paper why such a simple combination is suboptimal and provide an adaptive aggregation method with theoretical derivation, which is innovative. Furthermore, we expose and solve the problem of local overfitting when using BPT as local optimization in non-IID settings through input perturbation, which is also original and novel. In addition to reducing communication costs and supporting the clients to conduct optimization through inference only, a novel and effective aggregation with theoretical guidance is always thought of as a major contribution to the FL community. In addition, unveiling the optimization issue and improving the performance when applying BPT in non-IID settings is also a major contribution to FL.
>
> **W3**. We appreciate that more advanced baselines could be provided. However, we want to claim that outperforming the gradient-based methods in accuracy is not our goal. Our goal is to design a framework that enables resource-limited devices without model access to finetune LLM in a federated fashion. For example, when you are using ChatGPT, you do not have access to the model parameters and can only get the inference results by sending the request to the model through the API provided by OpenAI. Gradient-based methods can achieve relatively higher accuracy, but they are unfeasible in such realistic settings where the devices have no access to model parameters or have no power to conduct back-propagation. We conduct experiments primarily to show that we can improve accuracy compared with the other gradient-free methods with extremely efficient communication. Even though it is not our focus, we will include the discussion around these advanced gradient-based prompt tuning methods in our revision.
>
> **Q1**. A practical example of our setting is that the CMA-ES is deployed on the client’s device, and the LLM is deployed on an LLM service provider’s cloud (e.g., OpenAI). The clients conduct inference through the inference API (e.g., ChatGPT) and compute the evaluation results locally based on the inference results received from the API and the local ground-truth label. Thus, the clients do not need to have model access.
>
> **Q2**. As claimed above, we focus on realistic settings. Additionally to the limited access to the model parameters and constrained computational power, it is also practical that the clients have limited labeled data points and different data distributions, i.e., non-iid settings. Fine-tuning the whole LLM with a limited number of samples in FL causes model divergence across clients and cannot achieve good performance after aggregation. Thanks for this question, and we will include this discussion in the paper.

---

> > ### Comment · Reviewer_juc2 · 2023-11-23
> > **Thanks for the responses**
> >
> > Thank you for the rebuttal. Many of my concerns are not well addressed, such as w1, w3, and q2 and there are no new experiment results and no paper revision found. I would like to keep my rating.

---

### Official Review · Reviewer_zMti · 2023-11-04

**Soundness:** 3 good
**Presentation:** 3 good
**Contribution:** 2 fair
**Rating:** 5
**Confidence:** 4

**Summary:**

This paper proposed the FedBPT method which adopts CMA-ES to search for the optimal distribution of prompt and aggregates the collected local prompts in a sophisticated way. They also identified the overfitting issue caused by prompt tuning and applied the random masking techniques.

**Strengths:**

This paper is overall well-presented, and the motivation is also clear. The three challenges solved by FedBPT are verified by statements or experiments. The idea of getting the optimal step length seems good in the context of CMA-ES.

**Weaknesses:**

My primary concern pertains to the relatively limited contributions of this paper. It appears somewhat incremental when compared to BPT, despite the identification and resolution of certain issues, including a plain aggregation and overfitting.

**Questions:**

1.	I think the idea of applying BPT in FedBPT is direct and I have read the similar idea somewhere. So, it would be better to clarify the novelty and contributions of this submission?
2.	Authors claimed that standard aggregation is not effective in Sec 4.3, but I did not see any justification about this point except for Fig.2. Please let me know if I missed it.
3.	Following 1, standard aggregation does not need to send $F^k(z)$ to server if I have correctly understood, while it is necessary in FedBPT to get the optimal step length. In this case, the information about local data is at more risks of being exposed as more information shared out. A clarification is needed here.
4.	Sec 4.4 observed that most clients suffer from the dominant class issues. However, what is the loss for local clients when they are not well fitting across different classes? Also, is dominate class are “easy samples”?
5.	From experiments, the improvement of random masking seems quite limited. If the dominate class issue is sever as shown in Fig.3, the improvement are expected to be large.

---

> ### Author Response · Authors · 2023-11-16
> **Authors' response**
>
> **Q1**. Gradient-free methods were applied to finetune LLM prompts recently [1,2]. However, we did not find any existing work applying gradient-free optimization in FL for LLMs. We believe this is the first successful attempt in this field. We appreciate it if the reviewer could provide a reference if there are similar works. Additionally, our paper is not simply combining BPT and FedAvg. We show that simply combining BPT and FedAvg cannot achieve good performance, as shown in our results (the “FedAvg-BBT” baselines). We explore in our paper why such a simple combination is suboptimal and provide an adaptive aggregation method with theoretical derivation. Furthermore, we expose and solve the problem of local overfitting when using BPT as local optimization in non-IID settings. In addition to reducing communication costs and supporting the clients to conduct optimization through inference only, a novel and effective aggregation with theoretical guidance is always thought of as a major contribution to the FL community. In addition, unveiling the optimization issue and improving the performance when applying BPT in non-IID settings is also a main contribution to FL.
>
> **Q2**. Thanks for the comments. The reason why standard aggregation (e.g., FedAvg) is suboptimal when applying BPT is two-folded: 1. The clients locally optimize the CMA-ES parameters parameterized by multivariate normal distribution statistics. Mathematically, the arithmetic mean of the covariance matrices is not equivalent to the covariance matrix of our targeted optimal global distribution. Hence FedAvg is not effective under this setting. 2. CMA-ES, like other gradient-free methods, is a random search algorithm that cannot guarantee to achieve a local optimum as with gradient-based optimization algorithms. Therefore, directly averaging optimal and inferior local search results makes it difficult to achieve a global optimum. We will emphasize this more in our revision.
>
> **Q3**. F^k(z) represents the average loss value corresponding to the prompt z. Transmitting the average loss value will not cause serious privacy leakage of the data since the FL server has no access to the loss value corresponding to each sample. Transmitting local loss was also utilized in other FL works such as [3]. Thanks for this constructive suggestion, and we will include this clarification in our revision.
>
> **Q4**. We focus on the common FL setting rather than the personalized FL setting. Our goal is to train an optimal model with good performance on the global data distribution. In non-IID settings, the clients might have unbalanced training data distribution, but we should not train biased models for them since we cannot assume the data distribution after deployment. The samples of the dominant class are not “easy samples” within the global data distribution. The bias problem is caused by the unbalanced local training data in non-IID settings.
>
> **Q5**. We propose the masking technique to improve the accuracy of non-IID settings and mitigate the accuracy gap between the non-IID setting and the IID setting. The masking method reduces the accuracy gap between the non-IID setting and the IID setting by nearly 60% on SST-2 and 70% on AG’s News. Such results prove the effectiveness of masking in solving the accuracy degradation caused by non-IID.
>
> [1] Malladi, Sadhika, et al. "Fine-Tuning Language Models with Just Forward Passes." arXiv preprint arXiv:2305.17333 (2023).
>
> [2] Sun, Tianxiang, et al. "Black-box tuning for language-model-as-a-service." International Conference on Machine Learning. PMLR, 2022.
>
> [3] Tang, Minxue, et al. "FedCor: Correlation-based active client selection strategy for heterogeneous federated learning." Proceedings of the IEEE/CVF Conference on Computer Vision and Pattern Recognition. 2022.

---

> > ### Comment · Reviewer_zMti · 2023-11-23
> > **Thanks for your response**
> >
> > Thanks for the response.
> >
> > People in privacy community is sensitive to the extra information a model used. Thus, I strongly suggest theoretical analysis is needed when authors highlighted the privacy property.
> >
> > Regarding Q4, the authors explained that the bias problem is caused by unbalanced local data in non-IID settings. Since the all clients shared the same dominant class according to the paper, do you mean that the same class-imbalance happens across all local data?

---

### Official Review · Reviewer_2fPJ · 2023-11-04

**Soundness:** 2 fair
**Presentation:** 3 good
**Contribution:** 3 good
**Rating:** 6
**Confidence:** 4

**Summary:**

In this submission, the authors propose a novel federated learning (FL) framework named FedBPT to tackle the challenges when people want to fine-tune pre-trained language models (PLMs) via FL, including limited access to PLMs and unaffordable computation/storage/communication overhead. The proposed FedBPT utilizes black-box tunning (BBT) techniques to conduct the prompt tunning process, considering the prompt overfitting issue caused by non-IID data and the federated aggregation issues. Experiments are conducted on several benchmark datasets to show the effectiveness of FedBPT compared to both gradient-based baselines and gradient-free baselines.

**Strengths:**

1. The paper is well-written. The description of the proposed method is clear and detailed, making it easy to follow.
2. The authors highlight the challenges when directly using the existing BBT techniques in FL, and further provide solutions to solve these challenges. Some empirical evidence is provided in the experiment section to validate the effectiveness of the provided solutions.
3. The authors provide a theoretical analysis of the proposed method.
4. The experiments are comprehensive. The authors conduct experiments with RoBERTa and Llama 2, and provide comparisons from the perspectives of model performance, communication cost, memory, and so on.

**Weaknesses:**

1. The comparisons in terms of computation costs between the proposed method and baselines are not clear and convincing. Compared to gradient-based methods, FedBPT needs fewer computation resources per iteration since it adopts a gradient-free technique, but it is still not clear the overall computation costs used for obtaining the optimal solution. It is important since the gradient-free technique might need more iterations for convergence. The authors should provide more experiments or discussions.
2. The experimental results in Table 4 show that, even though 80% of the tokens are masked or replaced, the model trained on these ``noisy data'' outperforms that of vanilla BBT. The authors attribute such improvement to mitigating the overfitting issue. Such results are surprising (at least for me), and maybe the authors can provide more analysis or experiments. For example, the authors can replace FedAvg with other federated aggregation algorithms (e.g., FedProx[1]) that can also mitigate the overfitting issue caused by non-IID.

Refs:
[1] Federated Optimization in Heterogeneous Networks. In MLSys, 2020.

**Questions:**

Please refer to the Weaknesses above.

---

> ### Author Response · Authors · 2023-11-16
> **Authors' response**
>
> **W1**. Thanks for this constructive comment. Evolutionary algorithms are typically slower than SGD under the central training setting. However, in our design, each client plays the role of a “search node”, which improves the parallelism and accelerates the search process. To achieve the accuracies under non-IID settings of SST-2 in Table 1, FedPrompt requires more than 300 communication rounds, while our FedBPT only needs less than 100 communication rounds to converge. We will include an assessment of convergence time for all the settings and datasets in our revision.
>
> **W2**. As shown in Equation 7. Not only the noisy data is used, but also the clean data is used to evaluate the local prompt. Thus, utilizing samples with 80% of tokens masked does not mean that we lose 80% information of the samples. We want to find the prompts that make the LLM “confident” about the clean samples and “confused” about the noisy data. Thanks for the suggestion of applying FedProx. It is not straightforward to apply the proximal regularizer in FedProx to gradient-free optimization, and we will discuss this in the discussion section.

---

> > ### Comment · Reviewer_2fPJ · 2023-11-23
> > **Thanks for the responses**
> >
> > Thanks for the responses. I have read the authors' responses and other reviews, and tend to keep my rating.

---

### Official Review · Reviewer_aHU5 · 2023-11-05

**Soundness:** 3 good
**Presentation:** 3 good
**Contribution:** 2 fair
**Rating:** 6
**Confidence:** 3

**Summary:**

This paper introduces Federated Black-box Prompt Tuning (FedBPT) as a solution to tackle several challenges associated with the application of Federated Learning (FL) for fine-tuning Pre-trained Language Models (PLM). These challenges include issues related to parameter access, high computational demands, and communication overhead.
The proposed framework mitigates these challenges by minimizing the exchange of variables through gradient-free optimization known as Black-Box Tuning. This approach optimizes the prompt without relying on back-propagation. A series of experiments are conducted to verify the framework's capacity to reduce communication and memory requirements while still achieving competitive performance levels.

**Strengths:**

- The approach to address the overfitting problem is intriguing, particularly the use of random perturbation, which has proven effective in various domains.
- The paper's organization is well-structured. The problem formulation and inference presented in the paper are clear, making it accessible for readers.
- The techniques utilized within this framework demonstrate a level of innovation.

**Weaknesses:**

- Compared with gradient-based methods, FedBPT does significantly reduce communication overhead, but in the large-size LLama2 model, especially in the non_iid setting, its accuracy decline is still obvious.
- In the ablation study, the author states that FedBPT is not sensitive to the population. However,  three population trials are not enough to prove this parameter insensitivity in my view. More population trials should be performed.

**Questions:**

Q1: What are the rectangle colors in Figure 3? It should be better to give an explanation in the caption.
Q2: Random perturbation and controls the proportion of the perturbation are provided by generating a mask, my interesting is that the perturbation adding randomly or with semantic information? If this kind of perturbation is changed into words with opposite semantic information, for example, changing "zero" to "some" in Figure 4, can it have the same effect? Or produce other effects?
Q3: From the experimental results, it can be seen that the simple manual prompt also produces a comparable effect to FedAvg. Can the authors' approach can be improved by a prompt generated by FedBPT or selected from its dataset instead of manually prompting?

---

> ### Author Response · Authors · 2023-11-16
> **Authors' response**
>
> **W1**. Thanks for the appreciation of our improvement in communication efficiency. Our goal is to design a framework that enables resource-limited devices without model access to finetune LLMs in a federated fashion. For example, when you are using ChatGPT, you do not have access to the model parameters and can only get the inference results by sending the request to the model through the API provided by OpenAI. Gradient-based methods can achieve relatively higher accuracy, but they are unfeasible when the devices have no access to model parameters or have no computational power to conduct back-propagation. We conduct experiments primarily to show that we can improve accuracy compared with the other gradient-free methods with extremely efficient communication. For Llama 2, FedBPT improves accuracy by more than 12%, 11%, and 13% for SST-2, AG’s News, and Yelp compared with the manual prompts under non-IID settings, respectively. Such accuracy improvement is significant.
>
> **W2**. Thanks for this constructive comment. We will include experimental results by setting the population size as 40 and 100 in our revised version.
>
> **Q1**. We use a Viridis color map to represent the counts in the confusion matrix. The revised caption will be: “Figure 3: Confusion matrix of a client holding data that more than 90% is in class one. Each rectangle's color represents the number of counts, and darker shades indicate lower counts .”
>
> **Q2**. Our perturbation adds random information. It is interesting to change the words with opposite semantic information. In this case, the loss of the LLM inference results should be larger, which would boost the effect of our method formulated in Equation 7. We will leave it for future work.
>
> **Q3**. The method of selecting a prompt from the dataset is the In-Context Learning baseline in the experiments in which we randomly select up to 5 training samples (input text and labels)
> and concatenate them with the input texts. The improvement of our method is primarily from the optimized prompt generated by our method FedBPT. But we agree that if we initialize with the In-Context Learning prompt, there should be more improvement. Thanks for the constructive suggestion, and we will include this in our discussion section.

---

### Meta-Review · Area_Chair_ajir · 2023-12-06

**Metareview:**

This paper introduces the FedBPT method, leveraging CMA-ES to optimize prompt distribution and employing a sophisticated aggregation of locally collected prompts. The authors address the overfitting issue arising from prompt tuning by incorporating random masking techniques. Two reviewers underscore the impracticality of transferring data in federated learning, raising privacy concerns and questioning the feasibility of the proposed approach in real-world scenarios. The privacy community's concern about the additional information used by the model is emphasized, and the reviewers urge a thorough theoretical analysis to validate privacy claims. Additionally, critiques from two other reviewers center on the lack of novelty in modules of the paper, suggesting that limited technical contributions from relatively straightforward techniques. This concern is not adequately addressed in the author's rebuttal. In overall, the paper is deemed to fall below the expected standard of ICLR.

**Justification For Why Not Higher Score:**

N/A

**Justification For Why Not Lower Score:**

N/A

---

### Decision · Program_Chairs · 2024-01-16

Reject